# The Acute Effects of 25- Versus 60-Minute Naps on Agility and Vertical Jump Performance in Elite Youth Soccer Players: The Role of Individual Chronotype

**DOI:** 10.3390/life15030422

**Published:** 2025-03-07

**Authors:** Özgür Eken, Mertkan Öncü, Ahmet Kurtoğlu, Oguzhan Bozkurt, Musa Türkmen, Monira I. Aldhahi

**Affiliations:** 1Department of Physical Education and Sport Teaching, Faculty of Sports Sciences, Inonu University, Malatya 44280, Türkiye; ozgureken86@gmail.com; 2Department of Physical Education and Sport, Institute of Health Sciences, Inonu University, Malatya 44280, Türkiye; oguzhan.bozkurt@inonu.edu.tr (O.B.); musaturkmen4455@gmail.com (M.T.); 3Department of Coaching Education, Faculty of Sport Sciences, Bursa Uludag University, Bursa 16059, Türkiye; 4Department of Coaching, Faculty of Sport Science, Bandirma Onyedi Eylul University, Balikesir 10200, Türkiye; akurtoglu@bandirma.edu.tr; 5Department of Rehabilitation Sciences, College of Health and Rehabilitation Sciences, Princess Nourah bint Abdulrahman University, P.O. Box 84428, Riyadh 11671, Saudi Arabia

**Keywords:** chronotype, strategic napping, agility performance, youth football players, countermovement jump

## Abstract

Introduction: While napping is recognized as an effective strategy for mitigating insufficient sleep and enhancing athletic recovery, limited research exists on its effects on football players’ anaerobic performance, particularly concerning chronotype variations. This study investigated the impact of strategic napping durations on anaerobic performance and agility in football players under the age of 19 (U19), considering individual chronotypes and psychological factors. Methods: Sixteen young football players (age: 17.18 ± 1.04 years) participated in this crossover randomized controlled study. Participants underwent three conditions: no nap (NoN), 25 min nap (N25), and 60 min nap (N60), with 48 h washout periods between sessions. Performance was assessed using the Countermovement Jump Test (CMJ), Illinois Agility Test, and Illinois Change-of-Direction Test with Ball. Chronotype assessment, sleep quality, and mood states were evaluated using the Morningness-Eveningness Questionnaire, Pittsburgh Sleep Quality Index, and Profile of Mood States Questionnaire, respectively. Results: The 60 min nap protocol demonstrated significant improvements in agility performance compared to other conditions, particularly in the Illinois Agility Test and Change-of-Direction Test with Ball. However, no significant differences were observed in CMJ parameters across napping conditions. Chronotype variations showed correlations with agility performance and psychological factors, with evening-type participants displaying different responses to napping interventions compared to morning-type participants. Conclusions: While a 60 min post-lunch nap did not affect anaerobic performance, it positively influenced agility performance in soccer players. Chronotypic differences significantly impacted both agility performance and associated psychological factors. These findings suggest that integrating napping strategies into athletic training programs, while considering individual chronotypic variations, may present opportunities for enhancing specific aspects of athletic performance. Further research is needed to elucidate the underlying physiological, psychological, and cognitive mechanisms of these effects.

## 1. Introduction

Napping is considered an effective strategy for mitigating the short- and long-term negative effects of insufficient sleep, and due to its physiological and psychological restorative effects, it is regarded as an essential component of the recovery process, particularly for athletes [1,2]. For professional athletes, sleep not only supports recovery and restoration processes but also plays a crucial role in optimizing power output, making it a significant factor in performance [3]. Current recommendations suggest that healthy adults should aim for 7–9 h of sleep per night, while athletes may require a total of 9–10 h of sleep per day to meet their increased recovery needs [4,5,6]. Studies have shown that athletes tend to get even less sleep than the amount recommended for healthy adults, which can negatively impact both psychological and physiological performance [7]. Insufficient sleep has been attributed to various factors, including training schedules and load, light exposure, irritability, mood disorders, and increases in physical and mental stress [8,9]. As a result, there is growing interest in strategies to optimize sleep for athletes, with napping being considered a potentially beneficial tool for enhancing performance. It also provides an opportunity to supplement nighttime sleep, helping athletes achieve the recommended amount of rest [10,11].

Football players are exposed to physiological, metabolic, physical, and psychological stress factors resulting from training and competition-induced fatigue, which can lead to significant declines in their performance [12]. Various factors, such as night matches, travel, and intense scheduling, can disrupt normal sleep patterns, further exacerbating these stressors and negatively affecting recovery processes [13]. It has been demonstrated that post-exercise sleep restriction can have negative effects on perceptual recovery [14,15], cognitive functions [16], as well as aerobic [17] and anaerobic performance [18]. Furthermore, it has been reported that sleep restriction increases the risk of injury in football players [19]. Each of these factors significantly contributes to overall football performance [20].

However, it is important for physical therapists and trainers to consider inter-individual circadian frequency differences when planning training programs. While it is critical to understand how physical performance varies depending on the time of day, individual biological differences such as chronotype are also important factors to consider [21]. Chronotype refers to an individual’s predisposition to morning or evening and can be grouped as early chronotypes (ECTs), late chronotypes (LCTs), or intermediate chronotypes. Individuals with a morning chronotype generally wake up earlier in the morning, are more alert earlier in the day, and choose earlier bedtimes. On the other hand, individuals with an evening chronotype prefer later waking times, are more alert in the evening or at night, and have later bedtimes. While it is important to understand how physical performance varies depending on time of day, individual differences such as chronotype should also be taken into account [22,23]. Indeed, Brown et al. (2008) revealed in their study that rowing performance measured in the morning and evening hours was significantly better in the morning hours in athletes with an early chronotype [24]. However, a study by Rae et al. (2015) examined swimmers and found that athletes with a morning chronotype exhibited superior performance in morning trials, while those with an evening chronotype performed better in evening trials [25]. These studies reveal the importance of planning training and performance strategies in accordance with individuals’ chronotypes. In this context, the athlete’s chronotype and habitual sleep–wake time stand out as critical factors in associating sleep quality with athletic performance [3].

A growing body of literature supports the beneficial effects of napping on athletic performance [26,27]. In a study conducted by Hsouna et al. (2022), it was shown that a 40 min napping opportunity following an evening-simulated football match had positive effects on perceived effort levels [28]. Similarly, Ajjimaporn et al. (2020) found that a 20 min nap enhanced leg muscle strength recovery in male university football players [29]. By contributing to the limited information in the literature, this study aims to systematically investigate the effects of different nap durations on anaerobic performance. The hypothesis of this study is that strategic napping durations (25 min and 60 min naps) positively influence agility and countermovement jump performance in U19 football players, with the effects varying based on the duration and participants’ chronotypes.

## 2. Materials and Methods

### 2.1. Research Model

This research employs a crossover randomized controlled design to examine the impact of strategic napping on anaerobic performance in U19 football players, considering factors like chronotype, sleep habits, and sleep quality. Participants experienced three distinct conditions: no nap, a 25 min nap, and a 60 min nap. Each session was separated by a 48 h washout period to minimize carryover effects. This design enables the evaluation of different nap durations in a controlled and randomized environment, enhancing the reliability of the data by allowing participants to serve as their own controls.

### 2.2. Participants

To ensure scientific rigor and validate the collected data, a meticulous approach was used in selecting the study participants. The inclusion criteria were as follows: (a) being a football player in the Malatyaspor U19 team, (b) not having any illnesses or disabilities, and (c) not having a habit of napping. Exclusion criteria were strictly enforced to maintain the study’s internal validity and obtain reliable results. Participants were excluded if they (a) had active infections, (b) exhibited hyperactivity, (c) displayed unacceptable behavior, (d) did not comply with the study protocols, or (e) had difficulties following the study’s guidance. This rigorous application of criteria ensured a participant pool of football players without significant health issues that could impact the study results. It is important to note that none of the participants had a habit of napping.

The required sample size was calculated using G-power software 3.1.9.7 (University of Dusseldorf, Dusseldorf, Germany) [30]. Power analysis was conducted with F-tests (ANOVA: repeated measures, within factors) tailored to study design. Parameters for this analysis included an alpha error probability of 0.05, a minimum effect size of 0.65, and a power of 0.80 (1-β error probability), resulting in an actual power of 81.7% (Table 1). Based on this analysis, a minimum of 16 participants was deemed necessary. For a 10–15% attrition or inability to continue working, 19 people were included in the study. However, three athletes were excluded from the data analysis as they could not complete all required sessions. Consequently, the study proceeded with 16 young athletes. The participants’ ages ranged from 17.18 ± 1.04 years, with a height of 179.06 ± 3.17 cm, body mass of 71.5 ± 7.16 kg, body mass index of 22.28 ± 1.98 kg/m^2^, and resting heart rate values of 94 ± 14.38 beats per minute (Table 1).

This study adhered to the principles outlined in the Declaration of Helsinki. The study’s aims, rationale, and hypotheses were explained to the participants, and their informed consent was obtained. For participants under the age of 18, parental consent was also secured, as the participants were part of the U19 team. Additionally, necessary approvals were obtained from the Inonu University Non-Interventional Health Sciences Research Ethics Committee to ensure compliance with ethical principles (Ethics Committee Approval: 2024/6361, 15 October 2024), and it was conducted in accordance with the principles of the Declaration of Helsinki. This oversight ensured that the study was conducted according to established ethical standards and guidelines.

### 2.3. Experimental Design

Before the study commenced, three introductory sessions were held to acquaint the volunteer footballers with the established nap protocols and measurement procedures. The Self-Information Form was used to gather necessary personal information, the Morningness-Eveningness Questionnaire was employed to assess chrono-habits, the Pittsburgh Sleep Quality Index addressed sleep quality likely to influence the quality of strategic naps during the measurement periods, and the Profile of Mood States Questionnaire was utilized to observe mood states that may affect sleep quality. Following these preparations, participants attended the study site for three specific test sessions: no nap opportunity (NoN), a 25 min nap (N25), and a 60 min nap (N60), with a minimum of 48 h between sessions. Additionally, a balanced research design was used, with participants randomly assigned to follow three different strategic napping protocols (Table 2).

Upon arrival at the study site, volunteers were given ten minutes to adapt to the nap environment [31]. At 1:40 pm, volunteers chose their preferred lying position. Starting at 2:00 pm, the environment for the NoN, N25, and N60 protocols was provided in dark, quiet, and thermally appropriate sleeping rooms. Before all tests, it was accepted that the participants had taken eight hours of sleep according to their own declaration and did not drink any food or beverage other than water until at least 3 h before the measurements. During the nap sessions, participants in all conditions (NoN, N25, and N60) refrained from routine and visual activities such as using mobile phones and playing video games, as numerous studies have shown that visual activities can affect nap quality [32,33]. Anaerobic tests were administered 60 min after waking to alleviate the effects of sleep inertia [34]. After the strategic nap, participants completed a standardized warm-up routine, beginning with two minutes of light jogging, followed by three minutes of specific exercises, including foot crawls, toe and ankle rotations, trunk side stretches, trunk rotator stretches, hip circles, and knee bends. Heart rate variability was monitored using the Polar H10 in the resting position and during this warm-up protocol to avoid differences caused by warm-up intensity. Resting heart rates of the participants were analyzed in a sitting position after 10 min of passive rest before any exercise program. The Polar H10-controlled maximum heart rate warm-up protocol values averaged 144 ± 3.26 beats per minute. Following the warm-up, participants underwent anaerobic tests, including the Countermovement Jump Test, the Illinois Agility Test, and the Illinois Change-of-Direction Test with Ball (Figure 1). Nap application was performed at room temperature (20–22 degrees Celsius) in a sleep laboratory located in an indoor sports hall in the university campus, away from residential areas. The humidity in the room was kept between 40 and 60%. In order to minimize the noise; a day was chosen outside of class hours, when no one else was present except the participants.

### 2.4. Data Collection

#### 2.4.1. Anthropometric Measurements

The body measurements of the participants were taken using the SECA^®^ device (Gmbh, Hamburg, Germany), known for its precision in anthropometric assessments. To ensure consistency and accuracy, measurements were conducted under standardized conditions. Participants stood upright, barefoot, with their bodies, including ankles, calves, hips, buttocks, scapula, and head, aligned against a flat wall surface. This posture was essential for obtaining accurate body size data. The method followed the Frankfurt plane principle for determining head position, a standardized approach in anthropometry that ensures a repeatable and natural head posture during measurement. Each participant’s height was recorded during the inhalation phase, as this is the moment when the body is at its maximum extension, providing the most accurate height measurement. For body mass assessment, participants wore standardized lightweight clothing (Toledo 2096 PP, São Bernardo do Campo, Brazil) to minimize potential bias from heavier or bulkier clothing. Body Mass Index (BMI) was then calculated using a standardized formula, which involves dividing the participant’s weight in kilograms by the square of their height in meters [35]. This BMI calculation method is widely accepted in both clinical and research settings due to its simplicity and effectiveness in quickly assessing body fat distribution and potential health risks associated with various weight categories.

#### 2.4.2. Morningness-Eveningness Questionnaire

The Morningness-Eveningness Questionnaire (MEQ) is a widely used tool for assessing circadian typology. It includes 19 questions with Likert-scale responses regarding an individual’s preferred bedtimes, wake-up times, and activity periods. The answers are scored and totaled to generate an overall morningness score, which ranges from 16 to 86. Higher scores reflect a greater tendency toward morningness. Cronbach’s alpha for the reliability of MEQ was 0.812 and the test–retest reliability coefficient was 0.84 [36,37,38].

#### 2.4.3. The Pittsburgh Sleep Quality Index (PSQI)

In previous population-based studies focusing on sleep, the PSQI has been used because the PSQI provides important outputs in determining the sleep quality of individuals, and also because the results of the research are negatively affected when those with poor sleep quality are included in the research [39]. PSQI has a score range from 0 to 21, where 0 signifies no difficulty and 21 represents severe issues with sleep parameters [40]. This questionnaire consists of 19 self-assessment items covering seven distinct aspects of sleep: sleep quality, sleep duration, sleep latency, sleep efficiency, sleep disturbances, daytime dysfunction, and use of sleep medications. Additionally, there are five questions intended for spouses or roommates, but only the self-assessment responses were used in this study, as the roommate responses are primarily for clinical information. Each of the seven components is scored on a scale from 0 to 3. The overall PSQI score is calculated by summing the scores of these seven components, with higher scores indicating poorer sleep quality. Cronbach’s alpha for the reliability of the PSQI was 0.83 and the test–retest reliability coefficient was 0.82 [38].

#### 2.4.4. Profile of Mood States Questionnaire (PMSQ)

The PMSQ is designed to evaluate the emotional states of athletes. This tool assesses changes in mood by measuring emotions such as tension, depression, anger, fatigue, and confusion. Widely utilized in sports psychology, it helps understand athletes’ emotional conditions. The questionnaire features a 6-dimensional format with 58 questions. Athletes are asked to report their mood over the past week, including the day they complete the questionnaire, and their responses are scored. Scores range from 0 (Never) to 4 (Extreme). Mood can influence sleep quality and potentially diminish the benefits of strategic napping. In the analysis conducted to determine the validity and reliability of the PMSQ, it was concluded that the Cronbach coefficient was between 0.81 and 0.91 [41,42].

#### 2.4.5. Hooper Questionnaire

The Hooper questionnaire was utilized to evaluate signs of pre-fatigue and stress [43]. This tool included subjective assessments of sleep quality, fatigue, stress, and muscle soreness. Participants rated each factor using a 7-point Likert scale, where sleep ratings ranged from 1 (very, very good) to 7 (very, very bad), and ratings for fatigue, stress, and muscle soreness ranged from 1 (very, very low) to 7 (very, very high). These ratings were used to compute the Hooper Index (HI), which is the sum of the four ratings. The HI provides an overall measure of participants’ subjective experiences regarding sleep quality, fatigue, stress, and muscle soreness. A lower HI score indicates better well-being and greater readiness for training or competition, whereas a higher score suggests a possible decline in readiness or the presence of factors that could affect performance. In the validity and reliability studies conducted for the Hooper scale, Cronbach’s alpha coefficient was found to be 0.84 [44].

#### 2.4.6. Visual Analog Scale (VAS)

Subjective sleep quality was evaluated solely through self-report using a 10 cm VAS, ranging from “I Slept Very Poorly” to “I Slept Very Well.” VASs are favored for their simplicity, ease of administration, and suitability for self-completion. In the validity and reliability analyses performed for VAS scores, the test–retest validity of the scale was between 0.92 and 0.95, and the Cronbach’s alpha coefficient was between 0.93 and 0.96 [31,45].

#### 2.4.7. Countermovement Jump Test (CMJ)

Participants conducted each CMJ with their hands placed on their hips, beginning from a stationary standing position and keeping their legs straight during the flight phase of the jump. This specific posture was required to ensure consistency among participants and to reduce any external factors that might affect the jump’s result. The landing was executed with both feet maintaining ankle dorsiflexion. Participants were directed to jump as high as they could. Each participant was asked to complete a set of three CMJs, with a standardized one-minute rest period between each jump to allow for recovery and ensure consistent performance across trials. To ensure no impact on jump height, hand movement was restricted throughout the entire measurement process. Participants were instructed to begin the jump by quickly lowering their center of mass through knee flexion to a depth of their choice, followed by a maximal vertical leap from the lowest point achieved. For analysis, the highest of the three recorded jump heights for each participant was selected [46]. The My Jump Lab v.4.4.2 application by Dr. Carlos Balsalobre was used, which provides data on jump height, force, relative force, power, relative power, mean velocity, take-off velocity, impulse, and flight time.

#### 2.4.8. Illinois Agility Test

This test is known for its reliability and validity in assessing an athlete’s ability to rapidly change direction. It was performed within a 10-by-5 m area, marked with four cones placed in the center. Participants demonstrated their agility and speed by maneuvering around the cones [47].

#### 2.4.9. The Illinois Change-of-Direction Test with Ball

Participants undertook the Illinois Agility Test with Ball after completing the off-ball exercise to evaluate how anaerobic performance intersects with ball control skills. This sequence was designed to assess both the agility and speed of the participants while incorporating their ability to handle the ball effectively [48].

### 2.5. Statistical Analysis

In this study, statistical analyses were conducted using SPSS software (IBM, version 25, Chicago, IL, USA). To assess the normality of the data, the Shapiro–Wilk test was employed, confirming that the data followed a normal distribution. The Levene Test was used to evaluate the homogeneity of variances, which is crucial for validating the assumptions required for parametric tests like Analysis of Variance (ANOVA). Subsequently, a repeated-measures ANOVA was applied to examine the differences in CMJ, the Illinois Test, and the Illinois Change-of-Direction Test with Ball across the various napping conditions (NoN, N25, N60), with respect to chronotype. This type of ANOVA is particularly appropriate for designs where the same subjects are exposed to multiple conditions over time. Additionally, the Bonferroni post hoc test was utilized to identify specific differences between the tests. The results of the ANOVA were interpreted using the Mauchly Test of Sphericity; if the Mauchly Test yielded a value greater than 0.05, the assumption of sphericity was considered met. Otherwise, the Greenhouse–Geisser correction was applied. Repeated-measures correlation analysis of the relationship between repeated measures of napping was analyzed with R 4.2.5 software. (Auckland University, New Zealand) [49]. The effect size of the correlations was determined by considering the following thresholds [49,50]: <0.1 = trivial; 0.1–0.3 = small; >0.3–0.5 = moderate; >0.5–0.7 = large; >0.7–0.9 = very large; and >0.9 = nearly perfect. Effect sizes were calculated using Cohen’s d formula to assess the magnitude of the findings, with ANOVA effect sizes determined based on partial eta squared (ηp2) values. ηp2 values were interpreted as follows: ηp2 ≤ 0.01 indicated a small effect size, 0.01 ≤ ηp2 ≤ 0.06 indicated a medium effect size, and ηp2 ≥ 0.14 indicated a large effect size [51]. The significance level for the study was set at 0.05, and given the normal distribution of the data, results were presented as mean (M) and standard deviation (S.D.) [52].

## 3. Results

Figure 2 presents the comparison of CMJ parameters of the participants according to the duration of NAP. Accordingly, after NoN, N25, and N60, the following performance values were obtained: CMJ height [F_(1, 15)_ = 1.708, η_p_^2^ = 0.102, *p* = 0.198], force [F_(1, 15)_ = 1.298, η_p_^2^ = 0.080, *p* = 0.288], relative force [F_(1, 15)_ = 1.704, η_p_^2^ = 0.102, *p* = 0.199], power [F_(1, 15)_ = 1.686, η_p_^2^ = 0.101, *p* = 0.202], relative power [F_(1, 15)_ = 1.840, η_p_^2^ = 0.109, *p* = 0.176], mean velocity [F_(1, 15)_ = 1.476, η_p_^2^ = 0.090, *p* = 0.245], take-off velocity [F_(1, 15)_ = 1.436, ηp2 = 0.087, *p* = 0.254], impulse [F_(1, 15)_ = 1. 343, η_p_^2^ = 0.082, *p* = 0.276], flight duration [F_(1, 15)_ = 1.398, η_p_^2^ = 0.085, *p* = 0.263]. Accordingly, CMJ performance was not affected by nap duration.

Figure 3 presents the analysis of participants’ Illinois and COD test results based on nap duration. The results indicate that both the Illinois Test [F(1, 15) = 13.343, ηp^2^ = 0.406, *p* < 0.001] and the COD test [F(1, 15) = 10.239, ηp^2^ = 0.102, *p* = 0.002] showed significant differences over time. Post hoc Bonferroni analysis revealed that, for the Illinois Test, performance significantly differed between the NoN and N60 conditions [Δ = 0.846, *p* = 0.001, 95% CI: 0.33 to 1.35], as well as between the N25 and N60 conditions [Δ = 0.473, *p* = 0.023, 95% CI: 0.05 to 0.88]. Similarly, for the COD test, significant differences were observed between NoN and N60 [Δ = 1.634, *p* = 0.008, 95% CI: 0.41 to 2.85] and between N25 and N60 [Δ = 0.944, *p* = 0.005, 95% CI: 0.27 to 1.61]. These findings suggest that nap duration has a meaningful impact on agility and change-of-direction performance, with longer naps (60 min) leading to noticeable differences compared to shorter or no naps.

Figure 4 presents the analysis of participants’ VAS and Hooper Index test results based on nap duration. The results indicate that the Hooper Index showed some variation over time, but the effect was not statistically significant [F(1, 15) = 2.588, ηp^2^ = 0.147, *p* = 0.092]. Similarly, the VAS test results also varied over time but did not reach statistical significance [F(1, 15) = 0.966, ηp^2^ = 0.060, *p* = 0.341]. These findings suggest that while there were some changes in perceived fatigue and well-being based on nap duration, the differences were not substantial enough to be considered significant.

Table 3 presents the repeated-measures correlation analysis between participants’ MEQ scores and CMJ performance parameters. The analysis revealed that the CMJ performance parameters were highly correlated with each other (*p* < 0.001). Additionally, there was a weak positive correlation between MEQ scores and both RF-NoN (r = 0.25, *p* = 0.38) and RF-N60 (r = 0.24, *p* = 0.34). These results suggest that individuals with a stronger morning preference may have slightly higher relative force output in non-fatigued and fatigued conditions, though the correlations were low and not statistically significant (Figure 5).

Table 4 presents the repeated-measures correlation analysis between participants’ MEQ scores and their performance in the COD test, as well as their VAS and HI scores. The results indicate a negative correlation between MEQ scores and several variables, including COD-NoN (r = −0.24, *p* = 0.27) and COD-N60 (r = −0.30, *p* = 0.21), suggesting that individuals with higher morningness preference tend to perform worse in change-of-direction tasks. Additionally, significant negative correlations were observed between MEQ scores and HI-NoN (r = −0.55, *p* = 0.009), HI-N25 (r = −0.56, *p* = 0.005), and HI-N60 (r = −0.57, *p* = 0.002), indicating that those with a stronger morning preference reported higher perceived fatigue and stress levels under different conditions. These findings suggest that morning-type individuals may experience greater fatigue and decreased agility performance, particularly in scenarios involving different levels of physical exertion (Figure 6).

## 4. Discussion

This study is the first to investigate the impact of varying nap durations on the anaerobic performance and agility of football players, accounting for chronotype variations and psychological factors. The findings contribute significantly to the literature by extending the body of evidence in this area. Previous research has explored the influence of sleep-related interventions on physical performance across various sports, including karate [53], rugby [54], volleyball [55], judo and athletics [56,57], emphasizing its role as a valuable ergogenic aid. However, studies focusing specifically on the motor skills essential for football remain limited [12,28,29]. Addressing this gap, the present study evaluates the effect of different nap durations on anaerobic power and agility performance, while also considering individual chronotypic variations and psychological dimensions.

Analysis of the participants’ CMJ performance across different nap durations (NoN, N25, N60) revealed no significant differences in height, strength, relative strength, power, relative power, average speed, take-off speed, thrust, or flight time. These findings suggest that varying durations of strategic naps do not influence anaerobic performance. The results align with previous studies by Bentouati et al. and Daaloul et al., which reported that a 30 min midday nap did not enhance CMJ performance, a key indicator of anaerobic capacity [53,58]. Similarly, Petit et al. found that a 20 min midday nap had no effect on anaerobic performance following adequate nocturnal sleep [59]. Moreover, a recent meta-analysis by Boukhris et al. highlighted inconsistent findings regarding the impact of midday naps on anaerobic performance, further emphasizing the need for more research in this area [60].

However, the findings of this study contradict previous research suggesting that a 40 min nap enhanced short-term maximal performance reliant on anaerobic energy systems [28]. Similarly, other studies reported that a 20 min nap following sleep deprivation in football players led to partial improvements in minimal and average power during anaerobic sprint tests, while 20 min naps [61], and both 20 and 90 min naps were found to increase anaerobic power [62]. Vertical jump performance is a movement based on motor skills such as explosive power and muscle strength. This movement requires the muscles to contract synchronously and generate energy at the same time. Sleep is important for muscle recovery and replenishment of energy stores and the deep sleep stage in which slow-wave sleep is observed is of critical importance. However, 60 min sleep may not have sufficiently improved the explosive power and strength parameters required for vertical jump performance due to not containing enough of a deep sleep stage [63]. Research on afternoon naps has documented that slow-wave sleep exhibits a correlational relationship with sleep duration [63,64]. In a study they designed, Tanabe et al. reported that a 60 min afternoon nap contained 13.7 min of slow-wave sleep whereas a 90 min nap contained 16.0 min of slow-wave sleep [63]. Slow-wave sleep plays an important role in energy conservation which leads to more metabolic recovery [65]. In addition, growth hormone secretion increases during deep sleep and this hormone is important for muscle repair and growth [61]. While growth hormone is important for muscle repair and growth, 60 min sleep may not have contained enough of the deep sleep stage, so the secretion of this hormone may have been suppressed and, as a result, it may not have affected vertical jump performance.

Moreover, these discrepancies are likely due to differences in study protocols. Factors such as partial sleep deprivation [29,62], the specific performance tests used, sleep inertia [66], and varying nap durations may have contributed to the inconsistent results. As nap duration is a key determinant of athletic performance [67], its variability across studies could explain the conflicting outcomes. Moreover, the results of the present study suggest that chronotypic differences may play a role in the effectiveness of midday naps on athletic performance. A strong positive correlation was found between MEQ scores and CMJ performance. Specifically, CMJ performance improved with increasing morningness traits, as indicated by the positive correlations between MEQ scores and RF-NoN and RF-N60. However, since the CMJ tests were conducted in the afternoon, it is likely that participants’ chronotypic traits influenced their performance. These findings align with previous research showing that athletic performance is affected by chronotypic variations, with morning and evening chronotypes yielding different outcomes [68]. In summary, despite growing research, the effects of sleep-related interventions on anaerobic performance remain unclear, with varying results reported across studies. Further investigation is needed to clarify the conditions under which strategic napping may enhance performance.

Analysis of the impact of different nap durations on Illinois Test and Change of Direction (COD) test performance revealed that the N60 protocol resulted in superior outcomes compared to the NoN and N25 protocols in both tests. Napping practices influence various components of physical performance, including muscle strength, endurance, and anaerobic capacity. However, factors such as nap duration, sleep inertia, test protocols, and exercise types are critical determinants of these outcomes [60]. For instance, Botonis et al. (2021) noted in a review that most studies reported better performance outcomes following naps lasting 35–90 min compared to shorter naps of 20–30 min [67]. Similarly, Sirohi et al. (2022) emphasized that longer naps are more beneficial for optimizing athletic performance in physically active individuals [69].

In contrast, Bentouati et al. (2023) found that a 30 min nap did not improve performance in a karate-specific agility test, supporting the notion that nap duration is a crucial factor but also highlighting the potential for conflicting results between studies [58]. Extending the duration of midday naps may increase the proportion of slow-wave sleep, which plays a key role in both psychological recovery—such as reducing anxiety and stress—and physical recovery by mitigating muscle damage, thereby enhancing overall performance [34]. Slow-wave sleep during midday naps facilitates the restoration of peripheral and neural cells and contributes to energy conservation through increased parasympathetic activation. Furthermore, the positive relationship between prolonged slow-wave sleep and athletic performance suggests that the benefits of naps increase as the duration of slow-wave sleep episodes lengthens [67]. Agility, a physical performance parameter, is a more complex skill that involves different movement combinations such as changing place and direction, acceleration, and deceleration. Therefore, in addition to muscle strength, it also depends on neurological functions such as balance, coordination, reaction time, and decision-making. Sleep can make a significant contribution to the development of neurological functions [67]. Regardless of whether the sleep duration is short or long, REM sleep is also seen at different rates when individuals fall asleep [63,70]. During REM sleep, the levels of neurotransmitters such as serotonin and melatonin change [71]. This can affect psychological well-being sleep patterns and neurological functions. Therefore, a 60 min afternoon nap can improve neurological functions and increase agility performance. Because agility consists of complex movement combinations, it is more dependent on neurological parameters. This body of evidence may explain the favorable impact of the N60 protocol on Illinois and COD performance observed in the present study.

The findings of this study indicate that VAS scores were higher under the N60 protocol over time. Similarly, scores from the HI, which subjectively assesses variables such as pre-fatigue, stress symptoms, sleep quality, fatigue, stress, and muscle soreness, showed more favorable results in both the NoN and N60 protocols. Although the HI scores were highest in the NoN protocol, this difference was not statistically significant and did not translate into enhanced athletic performance. These outcomes may partially elucidate the positive impact of the N60 protocol on physical performance, as naps have been shown to alleviate negative psychological states such as stress and anxiety [34], which in turn may enhance mood and reduce sleepiness. This psychological benefit may further explain the superior Illinois Test and COD performance observed in the N60 protocol compared to the N25 protocol. In conclusion, factors such as nap duration, sleep inertia, and exercise type, alongside participants’ current psychological state, appear to be critical determinants in the effectiveness of nap strategies for athletic performance.

Other significant findings from this study include the negative correlations observed between MEQ scores and COD performances in both the no nap (COD-NoN) and N60 protocols, as well as between MEQ scores and HI scores in the no nap (HI-NoN), N25 (HI-N25), and N60 (HI-N60) protocols. An increase in the MEQ score indicates a greater morningness preference, which corresponds with a reduction in the mean duration of the COD-NoN and COD-N60 tests, suggesting improved performance in the COD test as morningness increases. Notably, the COD-N60 score was significantly higher than the COD-NoN score, indicating that the differences between these protocols may stem from the beneficial effects of strategic napping. Post-lunch sleep, irrespective of prior sleep history, serves as a crucial ergogenic aid, significantly enhancing psychological well-being, cognitive function, physical performance, and hormonal regulation for individuals [68,72,73,74]. Furthermore, these findings suggest that chronotypic variations may modulate agility performance, highlighting the potential role of individual circadian preferences in optimizing physical and cognitive outcomes. It was also observed that HI scores increased as MEQ scores decreased, suggesting that participants with an evening chronotype may be more susceptible to negative variables such as pre-fatigue, stress symptoms, sleep quality, fatigue, and muscle soreness. Investigating the potential effects of chronotypic differences on napping, Zamore et al. (2021) reported that 60 min of napping yielded more favorable outcomes and positively influenced emotional regulation in individuals with evening characteristics [75]. Similarly, Fang et al. (2019) examined the relationship between nap strategies, chronotype, and cognitive performance in elderly participants. They found that sleep durations exceeding 60 min were more effective in preserving cognitive functions in elderly individuals with morning characteristics, while no such association was evident in those with evening or intermediate chronotypes [76]. Individuals with different chronotypes may exhibit different physical [57,68], psychological, or cognitive performance [76] throughout the day. Individuals’ chronotype characteristics, such as being a morning type, evening type, or intermediate type, are decisive in exhibiting high performance [77]. A systematic review conducted by Vitale and Weydahl reported that morning-type individuals exhibited significantly higher performance in the morning hours compared to evening-type individuals [21]. Similarly, in a study conducted by Eken et al., morning-type judokas’ morning performance was found to be higher than evening-type judokas’ performance [68]. This body of evidence suggests that chronotypic characteristics exert psychological, physical, emotional, and cognitive effects on napping across different populations, potentially influencing the efficacy of midday napping.

This study was conducted with U19 male football players. Future research involving different populations or focusing on specific motor skills required by various sport disciplines may facilitate a more comprehensive analysis of the effects of napping. The participant imbalance between chronotypes in this study makes it difficult to compare the relevance of the results for each chronotype characteristic. Therefore, further research with a larger number of participants may have important implications for the generalisability of the results. An important limitation of our study is that nap durations were not adjusted according to the chronotype of the participants. In future studies, nap duration can be examined according to chronotype and physiological responses of different chronotypes to nap durations can be examined. An important limitation of our study is that the active falling asleep status of the participants during the sleep period was taken with their verbal feedback. In future studies, sleep status determination with actigraphy can be analyzed in depth.

## 5. Conclusions

This study demonstrates that a 60 min post-lunch nap does not have an effect on anaerobic performance; however, it positively influences agility performance in soccer players. Additionally, chronotypic differences appear to impact both agility performance and associated psychological factors. While these findings highlight the influence of chronotypic differences on napping interventions, further research is necessary to elucidate the potential physiological, psychological, and cognitive mechanisms underlying these differences. Integrating napping strategies into athletic training programs, with consideration of chronotypic variations, may present numerous opportunities for enhancing athletic performance.

## Figures and Tables

**Figure 1 life-15-00422-f001:**
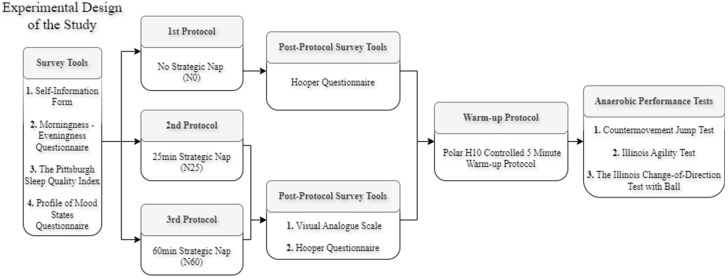
Experimental design of study.

**Figure 2 life-15-00422-f002:**
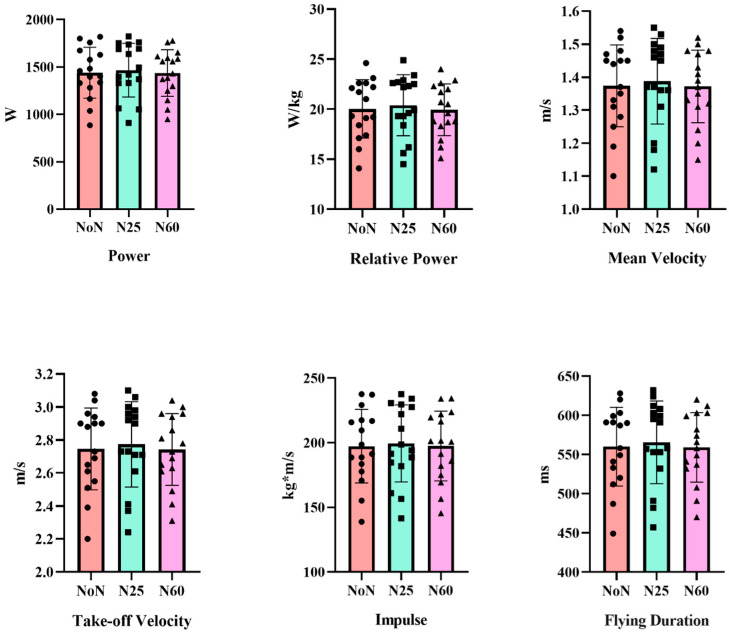
Comparison of CMJ parameters according to duration of nap.

**Figure 3 life-15-00422-f003:**
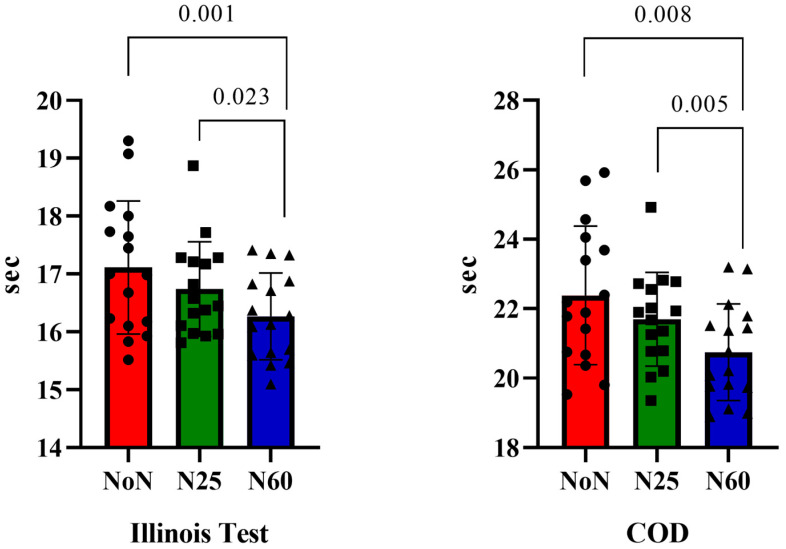
Comparison of Illinois and COD test result according to nap duration.

**Figure 4 life-15-00422-f004:**
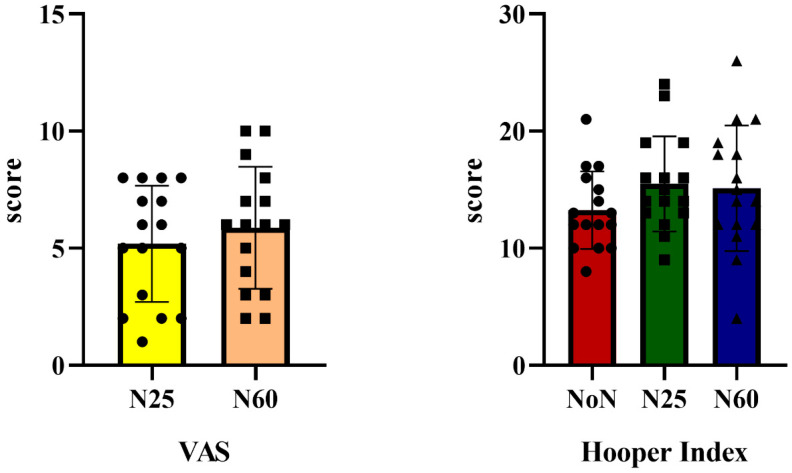
Comparison of VAS and Hooper Index results according to nap duration.

**Figure 5 life-15-00422-f005:**
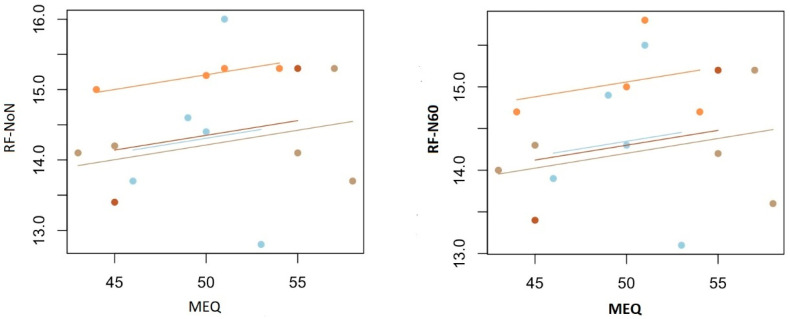
Repeated-measures correlation analysis between MEQ and RF.

**Figure 6 life-15-00422-f006:**
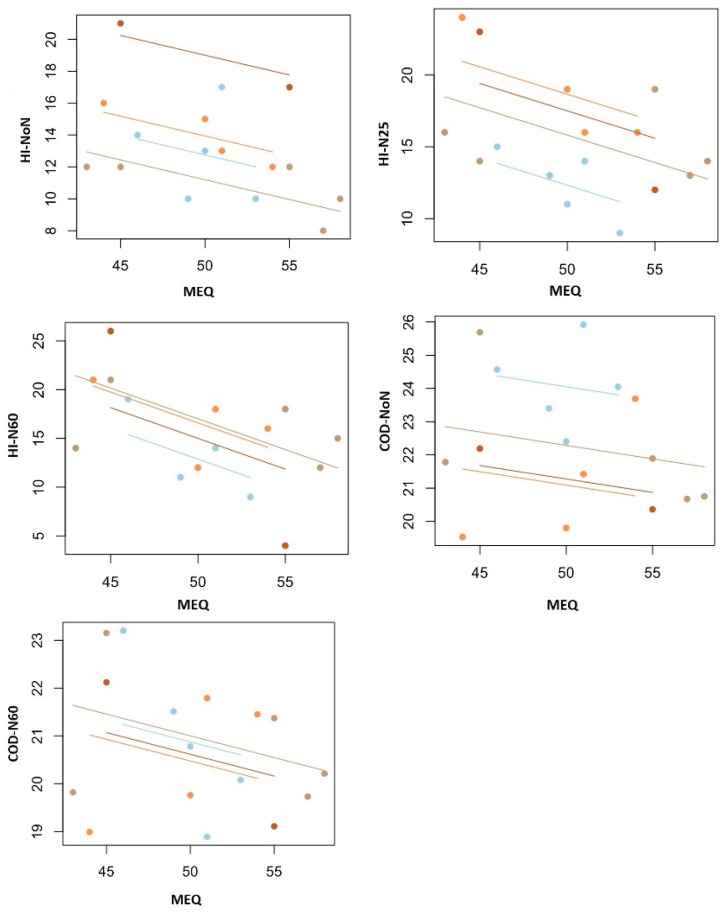
Repeated-measure correlation analysis results between MEQ score and HI and COD.

**Table 1 life-15-00422-t001:** Demographic information of participants.

Parameters	x ± SD	Min	Max
Age (year)	17.18 ± 1.04	16.0	19.0
Height (cm)	179.06 ± 3.17	173.0	183.0
Weight (kg)	71.50 ± 7.16	60.0	85.0
BMI (kg/m^2^)	22.28 ± 1.98	19.10	26.50
HR (beats)	94.00 ± 14.38	64.0	120.0
PSQI (score)	5.37 ± 2.62	2.0	13.0
PMSQ-Tension (score)	14.12 ± 12.24	6.0	26.0
PMSQ-Depression (score)	14.31 ± 12.24	0.0	38.0
PMSQ-Anger (score)	13.56 ± 9.68	2.0	29.0
PMSQ-Vigour (score)	16.81 ± 5.92	3.0	23.0
PMSQ-Fatigue (score)	8.75 ± 5.48	0.0	20.0
PMSQ-Confusion (score)	8.31 ± 4.85	0.0	16.0
PMSQ-Total Mood (score)	42.25 ± 35.10	−1.0	117.0
MEQ (score)	50.37 ± 4.75	43.0	58.0

HR: Heart Rate; PMSQ: Profile of Mood States Questionnaire; MEQ: Morningness-Eveningness Questionnaire.

**Table 2 life-15-00422-t002:** The order of implementation of the three exercise protocols in a balanced design among the 16 Participants.

Participants	1st Protocol	2nd Protocol	3rd Protocol
Participant 1	NoN	N25	N60
Participant 2	NoN	N25	N60
Participant 3	NoN	N25	N60
Participant 4	NoN	N60	N25
Participant 5	NoN	N60	N25
Participant 6	NoN	N60	N25
Participant 7	N25	NoN	N60
Participant 8	N25	NoN	N60
Participant 9	N25	NoN	N60
Participant 10	N25	N60	NoN
Participant 11	N25	N60	NoN
Participant 12	N60	NoN	N25
Participant 13	N60	NoN	N25
Participant 14	N60	N25	NoN
Participant 15	N60	N25	NoN
Participant 16	N60	N25	NoN

NoN: no strategic nap, N25: 25 min strategic nap, N60: 60 min strategic nap.

**Table 3 life-15-00422-t003:** Repeated-measures correlation analysis results between MEQ results and CMJ performance parameters.

	MEQ	JH1	JH2	JH3	F1	F2	F3	RF1	RF2	RF3	P1	P2	P3	RP1	RP2	RP3	TV1	TV2	TV3	I1	I2	I3	FT1	FT2	FT3
MEQ	1																								
JH1	0.12	1																							
JH2	−0.01	0.96	1																						
JH3	0.09	0.91	0.94	1																					
F1	0.08	0.61	0.62	0.54	1																				
F2	0.01	0.60	0.64	0.55	0.99	1																			
F3	0.02	0.55	0.60	0.54	0.98	0.99	1																		
RF1	0.25	0.96	0.90	0.87	0.54	0.52	0.48	1																	
RF2	0.12	0.95	0.96	0.90	0.57	0.58	0.53	0.96	1																
RF3	0.24	0.90	0.89	0.95	0.47	0.48	0.46	0.93	0.94	1															
P1	0.12	0.85	0.84	0.76	0.93	0.92	0.89	0.79	0.80	0.71	1														
P2	0.00	0.82	0.86	0.78	0.92	0.94	0.91	0.74	0.80	0.71	0.98	1													
P3	0.06	0.78	0.83	0.81	0.91	0.92	0.92	0.71	0.76	0.73	0.95	0.97	1												
RP1	0.18	0.99	0.94	0.90	0.59	0.57	0.52	0.98	0.96	0.91	0.83	0.79	0.75	1											
RP2	0.04	0.97	0.99	0.94	0.61	0.62	0.57	0.94	0.98	0.92	0.83	0.84	0.81	0.96	1										
RP3	0.15	0.92	0.93	0.99	0.51	0.53	0.51	0.90	0.93	0.98	0.74	0.75	0.79	0.91	0.94	1									
TV1	0.11	0.99	0.97	0.92	0.63	0.62	0.57	0.96	0.96	0.90	0.86	0.84	0.80	0.99	0.97	0.92	1								
TV2	−0.02	0.96	0.99	0.94	0.64	0.64	0.61	0.90	0.96	0.89	0.85	0.87	0.83	0.94	0.99	0.93	0.96	1							
TV3	0.09	0.92	0.95	0.99	0.56	0.58	0.56	0.87	0.91	0.95	0.78	0.80	0.82	0.90	0.94	0.99	0.93	0.94	1						
I1	0.03	0.71	0.72	0.64	0.98	0.98	0.96	0.62	0.65	0.55	0.96	0.96	0.94	0.67	0.70	0.61	0.72	0.74	0.66	1					
I2	−0.05	0.68	0.74	0.65	0.96	0.98	0.96	0.58	0.65	0.54	0.94	0.97	0.95	0.64	0.70	0.61	0.70	0.75	0.67	0.98	1				
I3	−0.02	0.64	0.69	0.65	0.96	0.97	0.98	0.53	0.60	0.54	0.91	0.94	0.96	0.59	0.66	0.61	0.65	0.71	0.67	0.97	0.98	1			
FT1	0.10	0.99	0.97	0.92	0.63	0.62	0.57	0.96	0.96	0.90	0.90	0.84	0.80	0.99	0.97	0.93	0.99	0.96	0.93	0.72	0.70	0.65	1		
FT2	−0.03	0.96	0.99	0.94	0.64	0.66	0.61	0.90	0.96	0.89	0.84	0.87	0.83	0.94	0.99	0.93	0.96	0.99	0.94	0.74	0.75	0.71	0.96	1	
FT3	0.07	0.92	0.95	0.99	0.56	0.58	0.56	0.87	0.91	0.95	0.78	0.80	0.83	0.90	0.94	0.99	0.92	0.95	0.99	0.66	0.67	0.67	0.82	0.83	1

MEQ: Morningness-Eveningness Questionnaire; JH: Jump Height; F: Force; RF: Relative Force; P: Power; RP: Relative Power; TV: Take-off Speed; I: Impulse; FT: Flying Time; 1 = NoN; 2 = N25; 3= NaN.

**Table 4 life-15-00422-t004:** Repeated-measures correlation analysis results of MEQ score and IT, COD, VAS, and HI results of participants.

	MEQ	IT1	IT2	IT3	COD1	COD2	COD3	VAS2	VAS3	HI1	HI2	HI3
MEQ	1											
IT1	−0.05	1										
IT2	−0.13	0.88	1									
IT3	−0.11	0.86	0.80	1								
COD1	−0.24	0.21	0.33	0.37	1							
COD2	−0.16	0.63	0.66	0.73	0.64	1						
COD3	−0.30	0.60	0.54	0.76	0.53	0.75	1					
VAS2	0.07	0.14	0.30	−0.11	0.08	0.18	−0.02	1				
VAS3	0.22	−0.21	−0.30	−0.30	−0.13	−0.20	−0.27	0.43	1			
HI1	−0.55	0.00	0.11	−0.05	0.25	0.10	−0.00	−0.02	−0.17	1		
HI2	−0.56	−0.08	−0.02	0.01	−0.01	−0.10	0.15	−0.50	−0.50	0.70	1	
HI3	−0.57	0.41	0.52	0.46	0.42	0.42	0.61	−0.19	−0.63	0.58	0.75	1

MEQ: Morningness-Eveningness Questionnaire, IT: Illinois Test, COD: Change of Direction, VAS: Visual Analog Scale, HI: Hooper Index, 1 = NoN, 2 = N25, 3 = NaN.

## Data Availability

The datasets generated and/or analyzed during the current research are available from the corresponding author on reasonable request.

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
