# Peer review of "The Acute Effects of 25- Versus 60-Minute Naps on Agility and Vertical Jump Performance in Elite Youth Soccer Players: The Role of Individual Chronotype"

_life, 2025, doi:10.3390/life15030422_

Round 1
Reviewer 1 Report
Comments and Suggestions for Authors
This study adopts a randomized crossover design to investigate the effects of nap duration on agility and psychological factors in U19 soccer players, which demonstrates a certain level of scientific rigor. However, the sample size consists of only 16 participants, which is relatively small and may not be sufficient to draw broadly applicable conclusions. It is recommended to increase the sample size to improve the statistical significance and generalizability of the results. Additionally, although the study considers differences in individual chronotypes, it does not clearly explain how these differences were grouped or addressed, resulting in a relatively weak analysis of the relationship between the results and circadian rhythms.
In lines 202–203, section 2.4.3 "The Pittsburgh Sleep Quality Index (PSQI)," the authors need to provide a rationale for using PSQI. I suggest adding the following sentence and reference at the beginning of this section: “Referring to previous population-based research focusing on sleep [doi: 10.1080/02640414.2024.2348906], PSQI was applied in this study. PSQI has a score range from 0 to 21, where 0 signifies no difficulty and 21 represents severe issues with sleep parameters (Buysse et al., 1989).”
In the results section, although the figures indicate improvements in Illinois and COD test performances, the lack of analysis on the potential physiological mechanisms behind the improvements observed with 60-minute naps weakens the support for the conclusions. The CMJ test results show no significant differences, yet the discussion section still compares these findings to other studies, which appears logically inconsistent. I recommend providing more detailed data analyses, exploring the reasons for the lack of significance, and supplementing with theoretical mechanisms and supporting literature.
The literature review covers the impact of napping on athletic performance but does not mention physiological mechanisms directly related to this study (e.g., neuromuscular recovery or metabolic effects). While the influence of chronotype is emphasized in the conclusion, there is insufficient experimental evidence to support its specific association with agility and psychological factors. It is suggested to include a more detailed discussion on the relationship between napping, agility, and psychological state, and to provide further analysis and references on the impact of chronotype.
Regarding methodological descriptions, the study lacks details on testing times and environmental conditions (e.g., temperature and noise control), which could impact the reliability of the results. Additionally, the reliability and validity of the scales used (e.g., MEQ and PMSQ) are not adequately explained. Supplementing validation details would enhance the study's credibility.
In the ethical statement, the authors mention obtaining informed consent from participants but do not explicitly state how the privacy of underage participants was protected. While the limitations section mentions sample homogeneity and the reliance on self-reported data, the discussion of the generalizability of the results is not sufficiently cautious and may overestimate the practical implications of the findings. A more comprehensive assessment of the study's limitations is recommended, along with revisions to address these issues.
In terms of language expression, the article contains repetitive statements and overly lengthy paragraphs, which affect readability. Figure captions are not intuitive enough, and it is suggested to use simpler titles and annotations to facilitate understanding. Refining the language and improving the overall quality of expression is recommended.
To sumup, this article holds certain research value but requires further improvement in sample size, data analysis, literature support, methodological details, and language expression. It is suggested that the authors revise the manuscript before resubmission. Wishing the authors success with the revisions.
Comments on the Quality of English LanguageModerate
Author Response
Dear reviewers,
Thank you very much for your valuable suggestions for our research. In this context, the corrections of the 1st reviewer are given in green, the corrections of the 2nd reviewer in yellow, and the corrections of the 3rd reviewer in blue in the text. We would like to thank the reviewers again for all their contributions.
Author's Reply to the Review Report (Reviewer 1)
Comment 1: In lines 202–203, section 2.4.3 "The Pittsburgh Sleep Quality Index (PSQI)," the authors need to provide a rationale for using PSQI. I suggest adding the following sentence and reference at the beginning of this section: “Referring to previous population-based research focusing on sleep [doi: 10.1080/02640414.2024.2348906], PSQI was applied in this study. PSQI has a score range from 0 to 21, where 0 signifies no difficulty and 21 represents severe issues with sleep parameters (Buysse et al., 1989).”
Response 1: The necessary correction has been made.
“In previous population-based studies focusing on sleep, the PSQI has been used because the PSQI provides important outputs in determining the sleep quality of individuals, and also because the results of the research are negatively affected when those with poor sleep quality are included in the research (You, 2024).”
Comment 2: In the results section, although the figures indicate improvements in Illinois and COD test performances, the lack of analysis on the potential physiological mechanisms behind the improvements observed with 60-minute naps weakens the support for the conclusions. The CMJ test results show no significant differences, yet the discussion section still compares these findings to other studies, which appears logically inconsistent. I recommend providing more detailed data analyses, exploring the reasons for the lack of significance, and supplementing with theoretical mechanisms and supporting literature.
Response 2: In line with your suggestions, the potential physiological mechanisms of the findings have been addressed and the scope of the discussion has been expanded. In this context, the following paragraph has been added to the discussion.
“Vertical jump performance is a movement based on motor skills such as explosive power and muscle strength. This movement requires the muscles to contract synchronously and generate energy at the same time. Sleep is important for muscle recovery and replenishment of energy stores and the deep sleep stage in which slow-wave sleep is observed is of critical importance. However, 60-minute sleep may not have sufficiently improved the explosive power and strength parameters required for vertical jump performance due to not containing enough deep sleep stage (Tanabe et al., 2020). Research on afternoon naps has documented that slow-wave sleep exhibits a correlational relationship with sleep duration (Brooks & Lack, 2006; Tanabe et al., 2020). In a study they designed, Tanabe et al. reported that a 60-minute afternoon nap contained 13.7 minutes of slow-wave sleep whereas a 90-minute nap contained 16.0 minutes of slow-wave sleep (Tanabe et al., 2020). Slow-wave sleep plays an important role in energy conservation which leads to more metabolic recovery (Shapiro et al., 1981). In addition, growth hormone secretion increases during deep sleep and this hormone is important for muscle repair and growth (O’Donnell et al., 2018). While growth hormone is important for muscle repair and growth 60-minute sleep may not have contained enough of the deep sleep stage so the secretion of this hormone may have been suppressed and as a result, it may not have affected vertical jump performance."
Comment 3: The literature review covers the impact of napping on athletic performance but does not mention physiological mechanisms directly related to this study (e.g., neuromuscular recovery or metabolic effects). While the influence of chronotype is emphasized in the conclusion, there is insufficient experimental evidence to support its specific association with agility and psychological factors. It is suggested to include a more detailed discussion on the relationship between napping, agility, and psychological state, and to provide further analysis and references on the impact of chronotype.
Response 3: In line with your suggestions, revisions have been made taking into account the existing literature. In this context, the following paragraphs have been added to the study.
“ Agility, a physical performance parameter is a more complex skill that involves different movement combinations such as changing place and direction, acceleration and deceleration. Therefore, in addition to muscle strength, it also depends on neurological functions such as balance, coordination, reaction time and decision-making. Sleep can make a significant contribution to the development of neurological functions (Botonis et al., 2021). Regardless of whether the sleep duration is short or long, REM sleep is also seen at different rates when individuals fall asleep (Petit et al., 2018; Tanabe et al., 2020). During REM sleep the levels of neurotransmitters such as serotonin and melatonin change (Holst & Landolt, 2022). This can affect psychological well-being sleep patterns and neurological functions. Therefore, a 60-minute afternoon nap can improve neurological functions and increase agility performance. Because agility consists of complex movement combinations it is more dependent on neurological parameters”.
Because post-lunch sleep, whether or not there is a normal sleep history, is an important ergogenic aid that significantly increases psychological well-being (Waterhouse et al., 2007), cognitive skills (Romdhani et al., 2021), physical performance (Eken, et al., 2024), and hormonal activities (Vgontzas et al., 2007) for individuals."
“Individuals with different chronotypes may exhibit different physical (Eken et al., 2024a; Kurtoğlu et al., 2024), psychological (Taylor & Hasler, 2018), or cognitive performance (Fang et al., 2019) throughout the day. Individuals chronotype characteristics such as being a morning type, evening type or intermediate type are decisive in exhibiting high performance (Montaruli et al., 2021). A systematic review conducted by Vitale and Weydahl reported that morning-type individuals exhibited significantly higher performance in the morning hours compared to evening-type individuals (J. A. Vitale & Weydahl, 2017). Similarly, in a study conducted by Eken et al., morning-type judokas morning performance was found to be higher than evening-type judokas (Eken, et al., 2024b)."
Comment 4: Regarding methodological descriptions, the study lacks details on testing times and environmental conditions (e.g., temperature and noise control), which could impact the reliability of the results. Additionally, the reliability and validity of the scales used (e.g., MEQ and PMSQ) are not adequately explained. Supplementing validation details would enhance the study's credibility.
Response 4: Nessesary sentence was added.
“NAP application was performed at room temperature (20-22 degrees Celsius) in a sleep laboratory located in an indoor sports hall in the university campus, away from residential areas. The humidity in the room was kept between 40-60%. In order to minimise the noise, a day was chosen outside of class hours, when no one else was present except the participants.”
Comment 5: In the ethical statement, the authors mention obtaining informed consent from participants but do not explicitly state how the privacy of underage participants was protected. While the limitations section mentions sample homogeneity and the reliance on self-reported data, the discussion of the generalizability of the results is not sufficiently cautious and may overestimate the practical implications of the findings. A more comprehensive assessment of the study's limitations is recommended, along with revisions to address these issues.
Response 5: Revised.
“This study was conducted with U19 male football players. Future research involving different populations or focusing on specific motor skills required by various sport disciplines may facilitate a more comprehensive analysis of the effects of napping. The participant imbalance between chronotypes in this study makes it difficult to compare the relevance of the results for each chronotype characteristic. Therefore, further research with a larger number of participants may have important implications for the generalisability of the results.”
Comment 6: In terms of language expression, the article contains repetitive statements and overly lengthy paragraphs, which affect readability. Figure captions are not intuitive enough, and it is suggested to use simpler titles and annotations to facilitate understanding. Refining the language and improving the overall quality of expression is recommended.
Response 6: Revised.

Reviewer 2 Report
Comments and Suggestions for Authors
Dear authors,
The work you present is very valuable since the performance of the players can be affected by several aspects and putting emphasis on rest is very important. You provide all the details of the research, however in the writing there are some aspects that should be improved. I will highlight them:
- In the summary, as you have written it in parts, the first point cannot be called "abstract", it should be "introduction".
- The objective of the study is written in two different ways. Which is the correct one?
- Table 1 that appears in the methods, is really part of the results, since you are providing data from your study. In addition, lines 116 and 117 indicate the same information that is already observed in the table. In addition, this table 1 appears before the inclusion and exclusion criteria. That is, it does not make sense for you to present the data of the footballers and then explain the criteria. - I consider that Table 2 should be part of point 2.3 Experimental design, since it explains how to distribute the footballers in the groups.
- In the results, all the figures and tables precede their explanations, and it should be the other way around. First the text and then the figures or tables.
- In Tables 3 and 4 the significance of the correlations does not appear, only the value of the correlation. Therefore, trying to read the table without knowing which correlations are significant is very difficult to read.
- In Table 4 the description of the acronyms does not appear.
- Finally, the results should be rewritten to make it easier to understand.
Regarding some doubts raised by the study:
- In the introduction they do not mention anything about the "chronotype", but instead they introduce it as a variable. The doubt is whether there is nothing in the literature that relates it and that is the reason for not having described it. But in that case, you should describe it as a factor that has not been taken into account so far and you think it is important.
- How much rest time was there between the anaerobic tests? Did they all follow the same order?
- According to figure 1, the VAS was only used after 2 nap protocols. What was the reason?
Thank you very much.
Author Response
Author's Reply to the Review Report (Reviewer 2)
Comments and Suggestions for Authors
Dear authors,
The work you present is very valuable since the performance of the players can be affected by several aspects and putting emphasis on rest is very important. You provide all the details of the research, however in the writing there are some aspects that should be improved. I will highlight them:
Comment 1: In the summary, as you have written it in parts, the first point cannot be called "abstract", it should be "introduction".
Response 1: Revised
Comment 2: The objective of the study is written in two different ways. Which is the correct one?
Response 2: Revised.
Comment 3: Table 1 that appears in the methods, is really part of the results, since you are providing data from your study. In addition, lines 116 and 117 indicate the same information that is already observed in the table. In addition, this table 1 appears before the inclusion and exclusion criteria. That is, it does not make sense for you to present the data of the footballers and then explain the criteria.
Response 3: Revised
Comment 4: I consider that Table 2 should be part of point 2.3 Experimental design, since it explains how to distribute the footballers in the groups.
Response 4: Revised.
Comment 5: In the results, all the figures and tables precede their explanations, and it should be the other way around. First the text and then the figures or tables.
Response 5: Revised.
Comment 6: In Tables 3 and 4 the significance of the correlations does not appear, only the value of the correlation. Therefore, trying to read the table without knowing which correlations are significant is very difficult to read.
Response 6: We wanted to write p values in the table, but the table was too long. For this reason, with the suggestions of the 3rd reviewer, we indicated the significant differences in bold. Because the CMJ parameters were performed with My-Jump application, there was a relationship between the sub-parameters. For this reason, instead of writing p-values for all of them, we only gave the ones with significant differences in the explanation, but we only bolded them to make the table more understandable.
Comments 7: In Table 4 the description of the acronyms does not appear.
Response 7: Revised.
Comment 8: Finally, the results should be rewritten to make it easier to understand.
Response 8: Revised.
Regarding some doubts raised by the study:
Comment 9: In the introduction they do not mention anything about the "chronotype", but instead they introduce it as a variable. The doubt is whether there is nothing in the literature that relates it and that is the reason for not having described it. But in that case, you should describe it as a factor that has not been taken into account so far and you think it is important.
Response 9: A paragraph added in “Introduction” section.
Comment 10: How much rest time was there between the anaerobic tests? Did they all follow the same order?
Response 10: The anaerobic performances of the participants were performed via CMJ performance and MyJump application. Therefore, the other parameters obtained are the parameters obtained from this test and the parameters obtained as a result of the formulas created by the system using the participants' weight, age, height, etc. as a result of a test. Therefore, there was no separate resting period for each of them. All anaerobic performance parameters were analysed in a single analysis.
Comment 11: According to figure 1, the VAS was only used after 2 nap protocols. What was the reason?
Response 11: In the study, VAS was asked to the participants to question the quality of sleep. Therefore, since the non-NAP group did not sleep, VAS questionnaire was not applied to them.

Reviewer 3 Report
Comments and Suggestions for Authors
Review for “Acute Effects of 25- versus 60-Minute Naps on Agility and Vertical Jump Performance in Elite Youth Soccer Players: The Role of Individual Chronotype”
Interesting work. The article is quite through from the statistical aspects. Please see below for my comments and questions.
Could you please include the MEQ information in Table 1?
Was there any care taken to adjust the time of the nap based on participants’ chronotypes as opposed to clock time? It is not clear if some of the subjects were from extreme chronotypes. If they were, then it would have been essential to set the nap times individually.
What is the reasoning for setting effect size to 0.65 in the a priori sample-size calculation? Would it be possible to perform a post-hoc power calculation?
In line#113 authors probably meant to say, “to account for an attrition rate of X%”, or something similar, instead of “To reduce the risk of participant dropout”.
How was resting heart defined and measured in this study? In general, the resting heart rate of athletes is much lower than 94 +/- 14 bpm which is the value reported here.
Were research participants able to take naps during the allocated time and for the specified time duration? As the participants were not habitual nappers, they might not have slept at all during the allotted time. Could you please clarify?
In the Hooper questionnaire, was the question regarding the quality of sleep used to qualify the nap, or the night-time sleep from the previous night? If this question was used for the nap, were subjects asked elsewhere about their quality of sleep from the night before the experiment? Regarding the fatigue component of this questionnaire, it would have been helpful to know the answer to this from before and after the nap, and for the pre-nap value to be used as a control. I understand that each subject is used as their own control, but this does not account for day-to-day variability within each subject. So ideally, the participants should have been asked about their level of fatigue before and after each nap.
In Tables 3 and 4, I suggest using bold face fonts for the correlation values that are statistically significant.
Author Response
Dear reviewers,
Thank you very much for your valuable suggestions for our research. In this context, the corrections of the 1st reviewer are given in green, the corrections of the 2nd reviewer in yellow, and the corrections of the 3rd reviewer in blue in the text. We would like to thank the reviewers again for all their contributions.
Author's Reply to the Review Report (Reviewer 3)
Comment 1: Could you please include the MEQ information in Table 1?
Response 1: Added.
Comment 2: Was there any care taken to adjust the time of the nap based on participants’ chronotypes as opposed to clock time? It is not clear if some of the subjects were from extreme chronotypes. If they were, then it would have been essential to set the nap times individually.
Response 2: In the study, NAP durations were not adjusted according to chronotype. A standard NAP programme was applied to all participants. Since there was not an equal number of participants from each chronotype, this was added to the ‘limitations’ section as a limitation of our study. However, this will be an important topic for future research. Thank you very much for your opinion in this context.
Comment 3: What is the reasoning for setting effect size to 0.65 in the a priori sample-size calculation? Would it be possible to perform a post-hoc power calculation?
Response 3: The effect sizes reported by previous studies examining similar variables generally range from medium to large (0.5-0.8). In our study, which we categorised according to Cohen's effect size, the estimated value is between 0.65, which is consistent with previous empirical findings. Poshoc power analysis was not performed in our study. According to the findings of previous studies, a power analysis was performed on average values.
Comment 4: In line#113 authors probably meant to say, “to account for an attrition rate of X%”, or something similar, instead of “To reduce the risk of participant dropout”.
Response 4: Revised.
Comment 5: How was resting heart defined and measured in this study? In general, the resting heart rate of athletes is much lower than 94 +/- 14 bpm which is the value reported here.
Response 5: In our study, we used HRs only for the effect of the warm-up protocol. Therefore, we did not establish an exclusion criterion for elevated baseline HRs. Since our study group consisted of individuals under the age of U19, sudden tachycardias may have occurred during the tests. However, this did not affect the results of our study. In this context, a sentence on how the resting heart rate was obtained was added.
“Heart rate variability was monitored using Polar H10 in the resting position and during this warm-up protocol to avoid differences caused by warm-up intensity. Resting heart rates of the participants were analysed in a sitting position after 10 minutes of passive rest before any exercise programme.”
Comment 6: Were research participants able to take naps during the allocated time and for the specified time duration? As the participants were not habitual nappers, they might not have slept at all during the allotted time. Could you please clarify?
Response 6: It was determined that the participants slept during this period with their own statements. The use of an actigraph in the article could have eliminated criticism of this situation, but no actigraph was used in our study. This may be a limitation of our study and we have mentioned this as a limitation of our study.
“An important limitation of our study is that the active falling asleep status of the participants during the sleep period was taken with their verbal feedback. In future studies, sleep status determination with actigraphy can be analyzed in depth.”
Comment 7: In the Hooper questionnaire, was the question regarding the quality of sleep used to qualify the nap, or the night-time sleep from the previous night? If this question was used for the nap, were subjects asked elsewhere about their quality of sleep from the night before the experiment? Regarding the fatigue component of this questionnaire, it would have been helpful to know the answer to this from before and after the nap, and for the pre-nap value to be used as a control. I understand that each subject is used as their own control, but this does not account for day-to-day variability within each subject. So ideally, the participants should have been asked about their level of fatigue before and after each nap.
Response 7: Hooper's questionnaire was applied to assess sleep quality during NAP in the study. Profile of Moods States Questionnaire (PMSQ) was used to determine their moods in the past week. The Pittsburgh Sleep Quality Index (PSQI) was used to assess the sleep quality of the previous night. Each participant reported that they had at least eight hours of sleep the previous evening and that they had not consumed food or drink other than water until at least 3 hours before the NAP. The PMSQ and PSQI questionnaires were used to exclude participants so that their results would not influence the NAP results. As you suggest, it would have been important to use the Hooper Survey before the NAP in order to see the initial situation. However, after the PMSQ and PSQI questionnaires, it was determined that all participants did not have any problems in terms of mood and sleep quality. Therefore, the Hooper questionnaire was not administered at baseline. However, this was an important information for our future researches, so we thank you again for your important suggestions.
Comment 8: In Tables 3 and 4, I suggest using bold face fonts for the correlation values that are statistically significant.
Response 8: Revised.

Round 2
Reviewer 1 Report
Comments and Suggestions for Authors
good study, good revisions, thanks for the authors' efforts.
Reviewer 2 Report
Comments and Suggestions for Authors
Dear authors,
Thank you very much for taking the comments suggested, the manuscript has improved considerably.
Congratulations on the work.
Best regards.